# Coverage-Based Summaries for RDF KBs

Giannis Vassiliou[1], Georgia Troullinou[2], Nikos Papadakis[1], Kostas Stefanidis[3],
Evangelia Pitoura[4], Haridimos Kondylakis[1,2]

[1] Department of Electrical and Computer Engineering, Hellenic Mediterranean University, He-raklion, Greece
[2] FORTH-ICS, Heraklion, Greece
[3]Tampere University, Tampere, Finland
[4]Computer Science and Engineering Department, University of Ioannina, Ioannina, Greece

giannis_vassiliou@yahoo.gr, troullin@ics.forth.gr,
npapadak@hmu.gr, konstantinos.stefanidis@tuni.fi,
pitoura@cs.uoi.gr, kondylak@ics.forth.gr

**Abstract.** As more and more data become available as linked data, the need for efficient and effective methods for their exploration becomes apparent. Semantic summaries try to extract meaning from data, while reducing its size. State of the art structural semantic summaries, focus primarily on the graph structure of the data, trying to maximize the summary's utility for query answering, i.e. the query coverage. In this poster paper, we present an algorithm, trying to maximize the aforementioned query coverage, using ideas borrowed from result diversification. The key idea of our algorithm is that, instead of focusing only to the "central" nodes, to push node selection also to the perimeter of the graph. Our experiments show the potential of our algorithm and demonstrate the considerable advantages gained for answering larger fragments of user queries.

## 1 Introduction

The rapid explosion of the available data in the web has led to an enormous amount of widely available RDF datasets. However, these datasets often have extremely complex and large schemas, which are difficult to comprehend, limiting the exploitation potential of the information they contain. One method for condensing and simplifying such datasets is through semantic summaries. According to our recent survey [1], a semantic summary is a compact information, extracted from the original RDF graph. Summarization aims at extracting meaning from data, and also at offering compact representations which some applications can exploit instead of the original graph to perform certain tasks.

State of the art works in the area of structural summarization [1], [2], first try to identify the most important nodes of the schema graph, and then to optimally link them, producing a connected schema sub-graph. As such, the size of the presented schema

graph is reduced to a minimum size, so that end-users are easier to understand the contents of the generated summary, while in parallel the most important nodes are selected and presented to the user.

**The problem.** The problem with the state-of-the-art structural semantic summaries is that the selected, most important nodes, are in most of the cases nodes located centrally to the graph, missing exploration opportunities for the nodes that are located at the perimeter of the graph. For example, consider the graph shown on the left of Fig. 1, which shows a summary from the state-of-the-art tool on structural summaries, i.e. the RDFDigest+ [2]. We can see that the summary focuses on a central part of the entire graph. Although such a summary would be really useful for queries around the *Agent* class (in the center), for a non-homogeneous query workload, a summary like the one presented on the right of Fig. 1 would arguably be better.

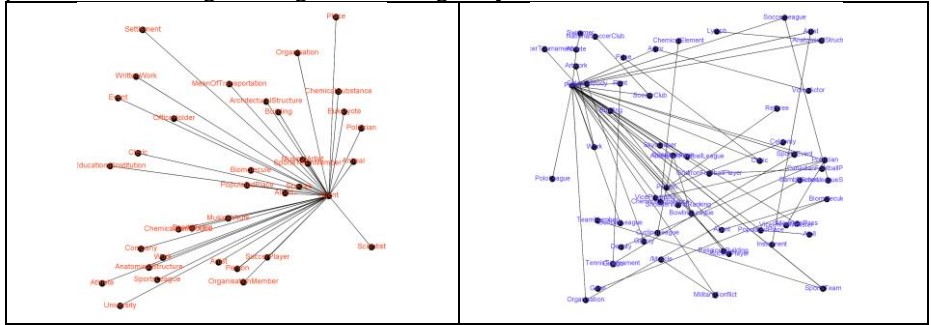

**Fig. 1.** Summaries for DBpedia from RDFDigest+ (left) and 1-LSP-Disc (right)

**Contribution.** To this direction, result diversification has also attracted considerable attention as a means of enhancing the quality of the exploration results presented to the users, as it offers, intuitively more informative results than a homogeneous result [3]. However, to the best of our knowledge, those ideas, although notably useful and interesting, have not yet migrated into structural semantic summaries. In this paper, we focus on summaries that try to maximize query coverage, exploiting ideas from the result diversification field. The idea is to combine semantic and structural diversity in order to further improve the generated summary, by first ordering the nodes based on their importance, and then iteratively starting from the nodes with the longest shortest paths, eliminate the nodes in the ranking within a specific radius, till the desired number of nodes is selected.

## 2    Schema Summarization

Schema summarization aims to highlight the most representative concepts of a schema, preserving important information and reducing the size and the complexity of the whole schema. Central questions to summarization are (i) how to select the schema nodes for generating the summary, and (ii) how to link selected nodes in order to produce a valid sub-schema graph [1]. To answer the first question, existing works so far exploit centrality measures, selecting the $k$ nodes with the highest value of the specific centrality

measure used (e.g. betweenness). To link those nodes, Graph Steiner-Tree [4] approximation algorithms are used, by introducing the minimum number of additional nodes to the summary - as introducing many additional nodes would shift the focus of the summary and decrease summary's quality.

In this work, we separate between the schema and instances of an RDF/S KB, represented in separate graphs, $G_S$ and $G_I$ as similarly done in the bibliography [1], [2]. The schema graph contains all classes and the properties they are associated with. The instance graph contains all individuals, and the corresponding properties. In our approach, we focus specifically on the schema graph and more specifically on *how to select the most important schema nodes so that summary's utility for query answering is maximized*. As such, assuming a query log, we would like to *maximize the fragments of queries that are answered by the summary*. More specifically, having a summary, we can calculate for each query that can be partially answered by the summary, the percentage of the classes and properties that are included in the summary, i.e. the success classes and the success properties. The *query coverage is the weighted sum of these percentages*.

Now, having defined the coverage for a given query workload $Q$, *a coverage-based summary is the one maximizing the coverage for the queries in $Q$*. As the problem of computing the summary with the maximum coverage for $Q$ is NP-complete, in this paper we propose a heuristic algorithm for computing it. The algorithm, named 1-LSP-DisC, starts by ranking all nodes based on the betweenness centrality measure (lines 3-5) and then it calculates all shortest paths for the top k/2 nodes in that list (lines 6-7). It selects the ones with the maximum distance and eliminates from the betweenness list their neighbors in a radius $r$. Then it continues visiting the remaining nodes in the list, removing each time, the neighbors of the selected nodes and so on (lines 9-15).

---

**Algorithm 1**: 1-LSP-DisC($G_S$,$k$, $r$)
**Input**: A graph $G$, $k$ the number of nodes to select, $r$ the radius of the nodes to be excluded.
**Output:** A set of nodes $N$.
1. $N := \emptyset$; $N_{LSP} := \emptyset$
3. **for each** *node* **in** $G_S$ **do**
4.     $betweenness[node] :=$ calculate_betweenness($G_S$)
5. sort_nodes(*betweenness*)
6. $N_{BETt} :=$ Select $k/2$ nodes from betweenness
7. $LSP :=$ Calculate_all_pairs_shortest_paths($N_{BET}$)
8. $N_{LSP} :=$ Select the two nodes from $LSP$ with maximum shortest path
9. **for each** *node* in $N_{LSP}$ **do**
10.     **Add** *node* to $N$
11.     **Remove** *node and node's* neighbors in a radius $r$ from the *betweenness* list
12. **while** *betweenness* $!= \emptyset$ and $|N| < k$ **do**
14.     **Add** *top node* in *betweenness* to $N$
15.     **Remove** *node* neighbors in a radius $r$ from the *betweenness* list
16. **Return** $N$

---

## 3    Evaluation & Conclusions

Next, we present a preliminary evaluation. We use as a baseline LSP and contrast our results with the state-of-the-art approach on structural summaries, the RDFDigest+ [2].

**LSP** selects k schema nodes with *the maximum shortest path distance to be included in the summary*, whereas **RDFDigest**+ selects the k schema nodes *with the highest betweenness value*. For **1-LSP-DisC**, we set the radius to one, as this is the only case were we could get in the summary the 10% of the available nodes (for r>1, many neighbors were excluded from the list and as such only few nodes were eventually left). In addition, as we are focusing on node-based summaries, for calculating coverage, we use 0.8 for the weight on classes and 0.2 for the weight on properties. For the evaluation, we use DBpedia v3.8 (422 classes, 1323 properties and more than 2.3M instances) and Semantic Web Dog Food (SWDF) KBs (120 classes, 72 properties and more than 300K triples) exploiting user query logs from the corresponding SPARQL endpoints (902 queries for SWDF and 56K queries for DBpedia). In each case, we request a 10% summary (16 nodes for SWDF and 36 for DBPedia).

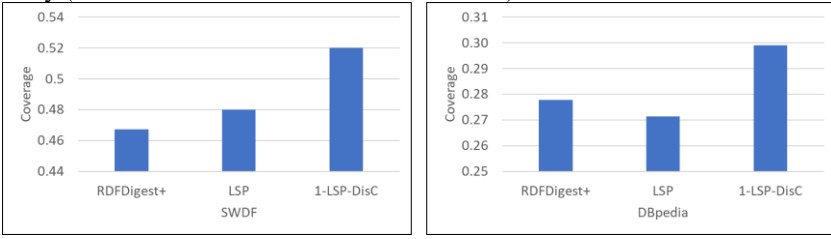

**Fig. 2.** Coverage for the various algorithms for the SWDF (left) and DBpedia (right) datasets.

As shown in Figure 2, in all cases out algorithm outperforms both LSP and RDFDigest+. In the case of SWDF, the RDFDigest+ generates a summary with a coverage of 47 %, the LSP achieves a coverage of 48% whereas the 1-LSP-DisC method achieves a coverage of 52%. The same is happening for DBpedia, where our approach achieves a coverage of 30%. Note that the difference in the coverage between the DBpedia and the SWDF dataset is attributed to the significantly larger size of DBpedia and to the large number of queries we had available for the DBpedia dataset.

Overall, our experiments confirm that the produced summaries indeed maximize the coverage of thousands of user queries, although they have been constructed, without using the specific workload. In our next steps, we intend to focus on properties selection, so that we not only select the ones minimizing addition of extra nodes in the summary, but also the most important ones, maximizing further the result coverage.

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
