# OpenReview forum: "Coverage-Based Summaries for RDF KBs"
_eswc-conferences.org/ESWC/2021/Conference/Poster_and_Demo_Track — ESWC2021 P&D_

### Official Review · AnonReviewer1 · 2021-04-08
**Research requires more discussion and formality**

**Rating:** 5
**Confidence:** 4

**Review:**

The authors present a new heuristic to summarise schema knowledge graphs. Although the problem and motivation are very clear and well documented by the authors, the contribution seems lacking formalisms, discussion and experiments.

The resolution of Fig. 1 could be improved, the names of the nodes are barely readable.

In Algorithm 1, there are several mistakes. The function calculate_betweenness requires both the graph and the node to compute the betweenness of the node in the graph. N_BETt then changes to N_BET. In line 6, it is not clear that the top k/2 nodes are selected, it gives the impression that any k/2 nodes are chosen. The function calculate_all_pairs_shortest_paths requires the graph as well, not just the selected nodes. All this indicate that the formalisation of the implementation can be improved.

In Section 3, it is required a longer discussion. What would happen if the weights of the coverage sum are changed? Why a summary of 10% of the nodes is good enough? How would impact the results if a different centrality measure is used? When discussing Fig. 2, it seems more appropriated to state that your heuristic performs best in "both" cases instead of "all" cases. Is a 4pp a good enough gain in coverage? Is the trade off with the execution time worth it? and with memory consumption? Without this discussion, I find it hard to know the actual value of such an heuristic.

**Anonymity:**

Yes, I would like my review to remain anonymous.

---

### Official Review · AnonReviewer4 · 2021-04-14
**A well written paper proposing an algorithm which outperforms the state of the art. A survey was performed previously and next steps are outlined. A promising contribution worthy to be presented and to receive feedback.**

**Rating:** 8
**Confidence:** 3

**Review:**

This poster paper proposes an algorithm to maximize query coverage by using result diversification, thus not only taking center nodes but also perimeter nodes into account.

## Summary

The problem is well defined, a relevant survey from the authors is linked which covers related work and other relevant papers are referenced in this poster.
Preliminary results are provided which show that the presented algorithm outperforms RDFDigest+ and LSP SWDF.
Future perspectives are provided.

Overall, this paper provides an interesting contribution.

Unclarities which should be resolved in case of acceptance:
for me it is not clear from the description in section 2 why exactly the schema and instance graph are separated.
Simply saying that this was "done in the bibliography [1], [2]" does not seem sufficient to me. The paper should be self-standing and at least half a sentence of the *why* should be provided.
Related to this, the algorithm description could be refined by specifying which graph is the input to the algorithm compared to the generic "a graph G".

Furthermore, in case of acceptance the extra-provided page should be used to explain the experimental setup and query logs, ideally by providing an example. This would help readers to understand the evaluation a bit better.

## Originality

Although the presented ideas are not novel itself, this work extends the state of the art by applying existing ideas in the field of graph summarization.

## Potential significance

Preliminary results indicate that state of the art is outperformed by the proposed algorithm.

## Topicality

The paper covers summarization of RDF graphs and thus is relevant to the conference.

## Clarity

The paper is well written and easy to follow.

## Minor

* the top frame of figure 1 overlaps with the text which appears that some text is underlined
* Paragraph below figure 2: "in all cases out algorithm" -> "our algorithm"

**Anonymity:**

Yes, I would like my review to remain anonymous.

---

### Official Review · ~David_Chaves-Fraga1 · 2021-04-14
**Interesting poster about RDF summarization lacking in comparison with other approaches**

**Rating:** 5
**Confidence:** 4

**Review:**

The poster presents an algorithm for summarizing RDF graphs based on the schema. It provides a description of the proposed and a preliminary set of results. Although the proposal seems interesting, the paper is not very easy to follow and should be improved, the figures are not very clear and should increase their size. Additionally, the introduction should at least, mentions previous approaches and what are their current issues, because neither the motivation nor necessity of the approach are clear.

Some recommendations:
1) Increase the size of figure one, divide it into two subfigures, and located it at the top of the page
2) Improve the description of the example provided in figure 1 (is it an example or a demonstration of the proposed contribution? it is confusing)
3) Figure 2 at the top and divide it into two subfigures. What is the main difference between your proposal and the previous one reported in the charts? Is an increase of 4% a significant improvement? why?
4) Use the same y labels and scales in figure 2.

Syntactic changes

In this paper --> In this poster, we present

**Anonymity:**

No, I would like my review to be deanonymized.

---

### Official Review · AnonReviewer3 · 2021-04-15
**The article presents an algorithm to compute summary graphs by combining semantic and structural features obtained from the entire graph. The article is well-written and structured. There is also a preliminar evaluation of the algorithm that shows promising results.**

**Rating:** 8
**Confidence:** 4

**Review:**

Quality: High

Clarity: High

Originality: High

Significance: High

Pros
- The article is well-written and structured.
- The information presented in the article is good to understand the idea behind de algorithm.
- The preliminar experimental results show that the algorithm outperforms the current solutions.

Cons
- The example presented in Section 1 is not adequate to understand the advantages of the proposed method. The authors should explain why the second summary graph presented in Figure 1 is better than the former.
- The description of the algorithm is too short. The authors should explain a little bit the novel features of the algorithms, e.g. the use of shortest paths for the top k/2 nodes with the best centrality.
- I can see that the coverage of the proposed algorithm can be hardly influenced by the target queries. So, it could be good to include some words with respect to such influence in the experimental evaluation.

**Anonymity:**

Yes, I would like my review to remain anonymous.

---

### Decision · Program_Chairs · 2021-04-19

Accept